# Treatment of High-Risk Neuroblastoma with Dinutuximab and Chemotherapy Administered in all Cycles of Induction

**DOI:** 10.3390/cancers15184609

**Published:** 2023-09-18

**Authors:** Margaret Cupit-Link, Sara M. Federico

**Affiliations:** St. Jude Children’s Research Hospital, Memphis, TN 38105, USA; sara.federico@stjude.org

**Keywords:** neuroblastoma, dinutuximab, chemoimmunotherapy, high-risk, Induction

## Abstract

**Simple Summary:**

The prognosis of high-risk neuroblastoma improved significantly with the addition of an anti-GD2 monoclonal antibody (dinutuximab) administered in the post-Consolidation setting. A subsequent study demonstrated improved efficacy when administering chemotherapy with dinutuximab to patients with relapsed or refractory neuroblastoma. We report the encouraging end of Induction response rates of a cohort of newly diagnosed high-risk neuroblastoma patients treated with concurrent chemotherapy and dinutuximab (chemoimmunotherapy) during all cycles of Induction therapy followed by Consolidation and post-Consolidation therapy.

**Abstract:**

Administration of chemoimmunotherapy using concurrent chemotherapy and an anti-GD2 monoclonal antibody (mAb), dinutuximab (DIN), demonstrated efficacy for the treatment of relapsed and refractory neuroblastoma. Chemoimmunotherapy, using a humanized anti-GD2 mAb, demonstrated a signal of activity in a phase 2 study for the treatment of patients with newly diagnosed high-risk neuroblastoma (HRNBL). In this single-institution retrospective study, patients with HRNBL received an Induction chemotherapy regimen plus DIN in all Induction cycles. Toxicity and response data were abstracted from the electronic medical record. Toxicities were graded by CTCAE v.5.0. The end of Induction (EOI) objective response rate was determined using the Revised International Neuroblastoma Response Criteria. Twenty-seven patients with HRNBL (23 newly diagnosed, 16 females, median age 3.9 years) started Induction chemoimmunotherapy from 27 January 2017 to 28 December 2022. All patients received DIN with all cycles of Induction therapy, and all but one patient completed Induction therapy. The most common non-hematologic grade ≥ 3 toxicities included fever (44%), hypoxemia (20%), and hypoalbuminemia (11%). End of Induction responses included eighteen with a complete response (CR), seven with a partial response (PR), one with progressive disease (PD), and zero with a minor response or stable disease. Twenty-six of twenty-seven patients (96%) completed all Induction cycles and were evaluable for a response. The EOI response of PR or better in the evaluable cohort was 96%. Dinutuximab was well tolerated with all Induction cycles, demonstrated an encouraging EOI response rate, and should be evaluated in a randomized study.

## 1. Introduction

Although significant progress has been made in the treatment of high-risk neuroblastoma (HRNBL), patient outcomes remain poor. The most recent randomized Children’s Oncology Group (COG) study of HRNBL, ANBL0532 (NCT00567567), demonstrated a 3-year event-free survival (EFS) of 51% [1]. In European trials HR-NBL1/SIOPEN and NB2004-HR, the 3-year EFS rates were 44–47 and 32–34%, respectively [2,3]. Thus, new therapeutic regimens are needed. 

In the relapsed/refractory NBL patient population, chemoimmunotherapy demonstrated significant efficacy in ANBL1221(NCT01767194), a two-arm pick the winner trial in which the patients who received irinotecan/temozolomide/DIN (I/T/DIN) had a superior response (53%) in comparison with those who received irinotecan/temozolomide/temsirolimus (6%) [4]. The chemoimmunotherapy arm was declared the early winner and an expansion cohort was added. The combined response rates of the initial and expanded cohorts demonstrated that the overall response rate of PR or better was 41.5% [4]. The significant results in the relapsed/refractory population provided the rationale for evaluation of chemoimmunotherapy in the upfront treatment of patients with newly diagnosed HRNBL.

Chemoimmunotherapy was evaluated in patients with newly diagnosed HRNBL in a single-institution phase 2 study, NB2012 (NCT01857934) [5]. In this clinical trial, newly diagnosed HRNBL patients received a novel humanized anti-GD2 monoclonal antibody (mAb), hu14.18K322A, in combination with a six-cycle Induction chemotherapy regimen [5]. The end of Induction (EOI) objective response rate ≥ PR was 93.7%, and the 3-year OS was 86% [5]. Following these results, patients treated at this center requested to be treated with Induction chemoimmunotherapy using the commercially available anti-GD2 mAb DIN in place of hu14.18K322A. Here, we report the toxicities and outcomes of a cohort of patients treated at a single center using a chemoimmunotherapy Induction regimen with DIN.

## 2. Materials and Methods

This is a single-institution retrospective review of 27 patients treated at St. Jude Children’s Research Hospital (SJCRH) for HRNBL with a chemoimmunotherapy Induction regimen using DIN in all Induction cycles. From 2017 to 2022, 67 patients with newly diagnosed HRNBL received treatment at SJCRH. Appendix A depicts a flow diagram of newly diagnosed HRNBL patients treated with different therapeutic regimens and treatment protocols. At the time of diagnosis, patients were invited to participate in any available clinical trial that was open at the time. Twenty-two patients were treated on NB2012, 13 on ANBL17P1, and 3 on ANBL1531. Two patients were treated as per ANBL0532 due to parental preference. The remaining 27 patients were treated “as per NB2012” using DIN in all cycles of Induction—the treatment plan described in this retrospective study. Patients included in this analysis were <19 years of age with histologically verified neuroblastoma categorized as HRNBL according to the International Neuroblastoma Risk Group Staging System (INRGSS) [6].

All patients received DIN with all cycles of Induction chemoimmunotherapy. Dinutuximab was given in the inpatient setting during days 2–5 of each cycle at a dose of 17.5 mg/m^2^ per day and on the same days as chemotherapy. Dinutuximab was infused over 10 h; however, in patients with hypersensitivity reactions, subsequent infusions were slowed to be given over 20 h. The DIN infusions were paused during infusions of chemotherapy but resumed immediately after. 

The first group of patients was treated as per NB2012, with 6 cycles of Induction chemoimmunotherapy using DIN in place of hu14.18K322A, along with Interleukin 2 (IL-2) and granulocyte-macrophage colony-stimulating factor (GMCSF). Then, they received Consolidation with single autologous stem cell transplantation (ASCT) using busfulan and melphan (Bu/Mel) conditioning, based on results of the HRNBL-1/SIOPEN study indicating that Bu/Mel conditioning led to superior outcomes compared with that of carboplatin, etoposide, and melphalan (CEM) [7]. After data from the HRNBL-1/SIOPEN study showed that use of IL-2 led to increased toxicities without benefit [8], IL-2 was removed from the treatment plan for subsequent patients. Later, tandem ASCT using thiopeta/cyclophosphamide (Thio/Cy) and CEM conditioning replaced single Bu/Mel ASCT, based on data from the ANBL0532 study [1]. As ANBL19P1 (NCT04385277) was an open COG study at the time of some patients’ post-Consolidation therapy, 3 patients were enrolled and received irinotecan and temozolomide (IRN/TMZ) along with isotretinoin and DIN during post-Consolidation. All other patients received post-Consolidation therapy as per ANBL0032 [9,10]. Most recently, patients were given 5 rather than 6 cycles of Induction chemoimmunotherapy, based on data from ANBL12P1 suggesting similar outcomes between groups of patients receiving 5 cycles versus 6 cycles [11]. All but 2 patients received surgical resection between cycles 3 and 5 at the discretion of the surgical team. One patient who did not undergo surgical resection had diffuse marrow disease at diagnosis without a primary tumor. The other had a previous resection during upfront treatment of intermediate neuroblastoma but did not undergo additional surgical resection because disease recurrence was in the bone marrow and skull only. A flow diagram depicting the variations in the treatment regimens that the patients in this study received is shown in Figure 1. Medications and dosages used in each cycle are shown in Appendix A.

Patient data were abstracted from the electronic medical record by a single investigator (M.C.-L.). Toxicities associated with chemoimmunotherapy administered during Induction were graded according to the National Cancer Institute Common Terminology Criteria for AEs (version 5.0) [12]. Only the highest grade was recorded for each patient per Induction cycle. Grades 1 and 2 toxicities were combined. Patients who received scheduled gabapentin and continuous narcotic infusions during DIN infusions were categorized as having grades 1–2 pain unless additional pain medications were added. All patients received scheduled diphenhydramine and acetaminophen as pre-medications for DIN. 

End of induction ORR was assessed using the Revised International Neuroblastoma Response Criteria (INRC) [13]. The evaluable population included 26 of 27 (96%) patients who had completed all planned Induction cycles. One patient was not included in the response assessment due to discontinuation of Induction therapy due to toxicity prior to completing EOI evaluations. The EOI ORR was dichotomized as ≥PR or <PR, and 95% CIs were calculated using the Clopper–Pearson method. End of therapy outcomes were calculated in the same way, although the 4 patients who had not completed therapy were excluded from the analysis. Descriptive statistics were used to provide summaries of patient characteristics and toxicities. 

## 3. Results

Twenty-seven patients, diagnosed between 2017–2022, received a chemoimmunotherapy Induction regimen outside of a clinical trial. The median age at diagnosis was 3.9 years (range: 0.3–10.7 years). Twenty-two patients (81%) were ≥18 months of age at diagnosis. Five patients (19%) were <1.5 years at the time of diagnosis but had *MYCN*-amplified disease, thereby qualifying them as having a high-risk disease. Eleven patients (41%) had *MYCN*-amplified disease. Eleven patients (41%) were male and 16 (59%) were female. All patients had stage M disease except for one patient with L2 *MYCN*-amplified disease. Sixteen patients (59%) were of white race, while seven (26%) were of black race, and four (15%) were classified as “other”. Although eight patients (30%) did not have histology classified based on the Shimada system [14], four (15%) had a favorable histology and fifteen (55%) had an unfavorable histology. Primary tumor sites included a single adrenal gland (88%), bilateral adrenal glands (4%), retroperitoneum (11%), and diffuse bone marrow disease without the presence of a primary tumor (4%). Sites of metastatic disease included bone (85%), bone marrow (78%), lymph nodes (44%), liver (15%), and lung or pleura (11%). Twelve patients had cytogenetic testing available to assess for the presence of segmental chromosomal aberrations. Of the total cohort, eight patients (30%) had 17q gain, two (7%) had 11q loss of heterozygosity (LOH), and none had 1p LOH. Twenty-three patients were newly diagnosed and had not received prior therapy. Three patients (two recurrent, one refractory) had previously received intermediate-risk therapy and subsequently required high-risk therapy for relapsed disease. One patient had recurrent disease after surgical resection and observation alone. Patient characteristics are shown in Appendix A. 

Twenty-six of twenty-seven patients completed all planned cycles of Induction; one patient discontinued therapy at the start of cycle 5 due to toxicity. This patient developed thrombotic microangiopathy (TMA) and diffuse alveolar hemorrhage (DAH) during Induction cycle 5; she is included in descriptions of patient characteristics and toxicities but not in the response analyses. The most common grade 3–4 toxicities were hematologic: febrile neutropenia (93%), anemia (100%), and thrombocytopenia (100%). The most common grade 3–4 nonhematologic toxicities were fever (4%), hypoxemia (20%), and hypoalbuminemia (11%). Other common toxicities possibly attributed to DIN included gastroparesis (19%), urinary retention (19%), and bladder spasms (30%). In nine patients (33%), DIN infusions were slowed to prevent infusion reactions. See Table 1 for toxicity data. At the EOI, eighteen of the twenty-six evaluable patients (69%) had a complete response (CR), seven (27%) had a partial response (PR), one had progressive disease (PD), and none had stable disease (SD) or death. One patient did not complete EOI evaluations due to toxicity and was not evaluable for response. The EOI ORR > PR in the evaluable cohort of 26 patients was 96%. The EOI responses are shown in Table 2.

At the time of data collection, twenty-two of the twenty-six evaluable patients had completed therapy (sixteen CR, two PR). Four patients were still receiving therapy. Four patients discontinued therapy due to progressive disease (four PD) that occurred during Induction (*n* = 1) or post-Consolidation (*n* = 3). Three patients died from disease, including two patients who experienced PD during therapy and one patient who had a CR at the end of therapy but developed a recurrent disease two months after completing therapy. At the time of data collection, seventeen patients had completed therapy without disease progression or recurrence. The median time off therapy for these seventeen patients is 9.8 months, ranging from 2.0 to 63.3 months. The end of therapy (EOT) responses are recorded in Table 2.

## 4. Discussion

Chemoimmunotherapy with DIN administered during all Induction cycles to patients with newly diagnosed HRNBL was tolerated and demonstrated a signal of activity. Only one of the twenty-six evaluable patients in our cohort experienced disease progression during Induction. All other patients responded to treatment, yielding a 96% EOI ORR of PR or better. The preliminary analysis of response at EOT was also encouraging, with 82% of the patients who had completed therapy having an ORR of PR or better. 

Although advancements in molecular diagnostics and treatment of neuroblastoma have been made over the past 2 decades, less is known about predictive markers of response. Pinto and colleagues performed a large retrospective review of 1242 patients treated on four consecutive Children’s Oncology Group (COG) HRNBL trials (A3973, ANBL02P1, ANBL0532, and ANBL12P1) [15]. In this analysis, an EOI response <PR was associated with a significantly lower EFS (3-year: 21% versus 54%; *p* < 0.0001) and OS (3-year: 46% versus 73%: *p* < 0.0001) [15]. In the reported cohort that received chemoimmunotherapy throughout Induction, all evaluable patients experienced an EOI of PR or better. While the mature long-term follow-up of this cohort has not yet been reached, the EOI response rate of PR or better may be an early signal of improved long-term outcome. 

Although the overall therapy delivered to patients in our cohort varied, there was minimal variability between the Induction regimens. The primary difference was that some patients received six cycles of Induction chemoimmunotherapy while others received five cycles. The optimal number of chemotherapy Induction cycles to maximize therapeutic effect in the treatment of patients with HRNBL is unknown and has not been evaluated in a randomized clinical trial. However, comparisons have been made between studies conducted in North America. Patients in the POG 9340 series of trials who received five cycles of Induction had similar EOI response rates to patients who received seven cycles of Induction therapy [16]. Further, at a single center, investigators compared the EOI CR and very good partial response (VGPR) rates between patients receiving five or seven cycles of Induction therapy [17]. In both study groups, the EOI response rates were approximately 80% [16,17]. In the COG HRNBL trial ANBL0532, patients received a six-cycle Induction regimen, and 80% of the patients who completed Induction therapy had an EOI response of PR or better [1]. In ANBL12P1, 80% of patients treated with a five-cycle Induction regimen had an EOI response of PR or better [11]. Thus, the variability of five versus six cycles of Induction therapy administered to the patients evaluated in this cohort likely had a negligible to minimal effect on the EOI response rates.

There were additional differences in the treatments received by the patients included in the reported cohort. These therapeutic changes were made in response to new emerging data. For example, when the results of the HR-NBL1/SIOPEN trial showed that the addition of IL-2 to post-Consolidation DIN-beta increased toxicity without a survival benefit [18], IL-2 was removed from the treatment plan. Additionally, when the results of ANBL0532 demonstrated that patients who received tandem ASCT had superior outcomes to those who received single ASCT [1], patients in the cohort were then treated with tandem ASCT. However, these changes did not impact the primary objectives of this study, which were to assess chemoimmunotherapy tolerability in newly diagnosed patients receiving DIN in all Induction cycles and to assess the EOI response rates. 

This study has several limitations. First, it is a single-center retrospective cohort study in which patients were treated at an institution with expertise in anti-GD2 immunotherapy administration to patients with a bulky disease. Therefore, the ability to administer DIN with all cycles of Induction therapy may be related to experienced staff proactively managing side effects and may not be generalizable to all settings. Specifically, symptomatic newly diagnosed patients with bulky disease may not be able to tolerate the initial cycle of chemoimmunotherapy if administered in a hospital with less experience. Although we chose to administer DIN in the first cycle of Induction, based on clinical experience related to the preceding clinical trial (NB2012), administration of anti-GD2 mAb therapy in Induction cycle 1 would need to be carefully assessed in a multi-institutional manner before feasibility could be determined. Second, most patients included in this cohort have been off therapy for a limited amount of time. Therefore, there is a paucity of long-term follow-up data. Third, this was not a randomized study. As shown in Appendix A, this cohort includes 27 of the 67 (40%) of the patients treated at SJCRH for newly diagnosed HRNBL from 2017–2022. Patients were treated with this non-protocol treatment plan based on clinical trial availability and parental choice at the time of diagnosis; so, selection bias may have impacted our results. Finally, a potential confounding variable in this cohort may be related to access to resources. All patients who received this therapy were provided housing, food, and transportation throughout the duration of therapy. This may have impacted results, as a study evaluating 371 patients with high-risk neuroblastoma treated with an anti-GD2 mAb recently demonstrated that poverty is independently associated with an increased risk of relapse and death among neuroblastoma patients treated with targeted immunotherapy [19].

Nonetheless, the signal of early clinical activity related to a chemoimmunotherapy Induction regimen is promising and warrants further investigation. The initial single-institution phase 2 study, NB2012 (NCT01857934), using a novel humanized antibody hu14.18K322A, had an encouraging EOI response rate that translated to an improved 3-year EFS and OS when compared with that of historical controls [5]. If chemoimmunotherapy using DIN during Induction therapy also leads to improved patient outcome, this has the potential to change the treatment paradigm, if validated in a randomized study, for patients with newly diagnosed HRNBL. 

Modifications of anti-GD2 mAb regimens and development of new combinations of therapy are ongoing and may further advance the field. Historically, numerous anti-GD2 mAbs have been evaluated for clinical use in neuroblastoma cases including 3F8, humanized 3F8 (naxitamab), 14.G2a, DIN, hu14.18K322A, humanized 14.18 fused with interleukin-2, and dinutuximab beta [20,21,22]. Some of these mAbs have been combined with other therapeutics, such as lenalidomide [23], vorinostat [24], and other agents, while others have altered dosing schedules and infusion durations. Optimization is ongoing, and both basic and clinical research continue to assess the most effective way to augment immunotherapy to achieve the most significant clinical benefit for this patient population. Future clinical trials seek to enhance antibody-dependent cell-mediated cytotoxicity (ADCC), the primary mechanism of action of the therapy, to improve efficacy.

While chemoimmunotherapy has been successful for improving response rates in patients with HRNBL, this represents only a subset of therapeutic strategies that may change the treatment paradigm in the future. In addition to anti-GD2 mAbs, other types of immunotherapies are being evaluated for the treatment of HRNBL. Chimeric antigen receptor (CAR)-expressive T cells targeting L1 cell adhesion molecules (CD171) have been studied [25,26]. Although responses were limited, more recent clinical trials assessing alternative CAR T-cell therapies have demonstrated a signal of activity. For the first time, administration of CAR T-cells targeting GD-2 demonstrated a clinical response in a subset of patients with relapsed/refractory HRNBL who were treated in a phase 1/2 study [27]. While this marks a critical inaugural step in what is hoped to be a future therapeutic option for a larger patient population, it is important to note that the patients who responded in the study included those with minimal disease. Patients with more significant tumor burden had a limited therapeutic benefit. In a different phase 1 study, administration of autologous natural killer T cells co-expressing a GD2-specific CAR with interleukin 15 was feasible and tolerated by children with relapsed or refractory NBL [28]. A phase 2 study to assess efficacy is ongoing. In the future, modifications to the constructs and targets of the next generation of CAR T-cell therapies may lead to more significant, durable responses in patients with a larger disease burden and be included in a future treatment paradigm.

In addition to immunotherapy, incorporation of other targeted agents may also alter future therapy for HRNBL. The phase 3 study, ANBL1531 (NCT03126916), is evaluating whether inclusion of the ALK inhibitor lorlatinib to the standard COG backbone in patients with *ALK* aberrant disease improves the outcome. Inclusion of this agent is supported by data from a phase 1 study (NANT 1502, NCT03107988) in which patients with *ALK* aberrant recurrent or refractory neuroblastoma tolerated the combination of lorlatinib with chemotherapy and demonstrated a clinical response [29]. ANBL1531 is also evaluating whether the addition of I131-metaiodobenzylguanidine (^131^I-MIBG) to the COG Induction chemotherapy backbone will improve the outcome for the 90% of patients with HRNBL who have a MIBG-avid disease. 

In summary, advances in therapies for the treatment of HRNBL suggest that there is potential to alter the HRNBL treatment paradigm in the future. If chemoimmunotherapy administered during Induction therapy, as reported in this cohort, demonstrates improved outcome in a randomized phase 3 trial, then future studies may alter other components of the treatment. In addition to improving the outcome, this could lead to the removal of the more toxic aspects of HRNBL therapy, including high-dose chemotherapy with stem cell rescue, and ultimately decrease late effects related to therapy.

## 5. Conclusions

Treatment of patients with HRNBL with DIN chemoimmunotherapy in all cycles of Induction is feasible and tolerable. The EOI response rates to this chemoimmunotherapy Induction regimen are promising. Assessment of Induction chemoimmunotherapy in a randomized clinical trial is warranted and has the potential to change the standard of care and outcomes for children with HRNBL.

## Figures and Tables

**Figure 1 cancers-15-04609-f001:**
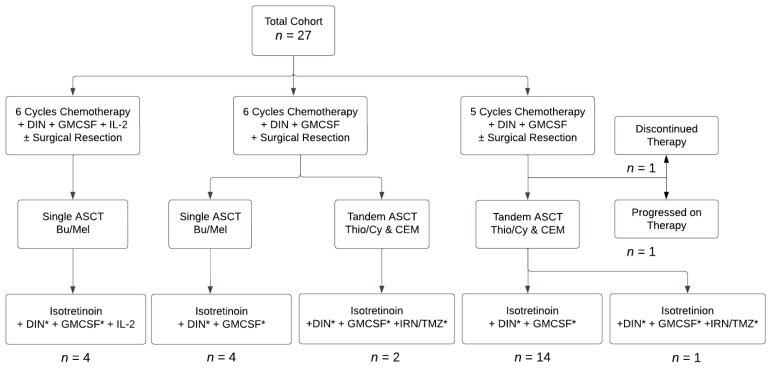
Flow diagram showing treatment patients received. Abbreviations: dinutuximab (DIN), granulocyte-macrophage colony-stimulating factor (GM-CSF), interleukin-2 (IL-2), autologous stem cell transplantation (ASCT), busulfan/melphalan (Bu/Mel), thiotepa/cyclophosphamide (Thio/Cy), cisplatin/etoposide/melphalan (CEM), irinotecan (IRN), temozolomide (TMZ). * Omitted in Cycle 6.

**Table 1 cancers-15-04609-t001:** Treatment-related toxicities (highest grade received) during chemoimmunotherapy Induction. Toxicities graded according to CTCAE version 5 [12] (highest grade for each patient).

Toxicity	Grades 1–2: *n* (%)	Grade 3: *n* (%)	Grade 4: *n* (%)	n/a
Bacteremia	7 (26)	n/a	n/a	
Sepsis	8 (30)	2 (7)	1 (4)	
Infusion-related				
Hypotension	8 (30)			
Bronchospasm	9 (33)	1 (4)	1 (4)	
Hypoxemia		5 (19)	1 (4)	
Allergic Reactions		1		
Pain	27 (100)			
Fever	13 (48)	12 (44)		
Hematologic				
Febrile Neutropenia		25 (93)		
Anemia		27 (100)		
Thrombocytopenia			27 (100)	
Prolonged Thrombocytopenia				1 (4)
Bone Marrow Hypocellularity			1 (4)	
Cardiopulmonary				
Pulmonary Hypertension	1 (4)			
Diastolic Dysfunction	1 (4)			
Hypertension	13 (48)	1 (4)		
Metabolic				
Hypokalemia	18 (67)			
Hypophosphatemia	25 (93)	2 (7)		
Hypocalcemia	6 (22)	1 (4)		
Hypoalbuminemia	24 (89)	3 (11)		
Hyponatremia	4 (15)			
Hypomagnesemia	7 (26)			
GI/GU				
Clostridium Difficile				12 (44)
Gastroparesis	3 (11)	1 (4)	1 (4)	
Urinary Retention	5 (19)			
Bladder Spasms	8 (30)			
Other				
Thrombotic Microangiopathy			1 (4)	
Diffuse Alveolar Hemorrhage			1 (4)	

**Table 2 cancers-15-04609-t002:** Disease response at end of Induction and end of therapy. Abbreviations: CR: complete response, PR: partial response, SD: stable disease, PD: progressive disease, CI: confidence intervals.

End of Induction Objective Response Rate	*n* = 26 (%)
CR	18 (69)
PR	7 (27)
≥PR	25 (96 [95% CI: 81, 99])
SD	0
PD	1 (4)
End of Therapy Response	*n* = 22 (%)
CR	16 (73)
PR	2 (9)
≥PR	18 (82 [95% CI: 61, 93])
SD	0
PD	4 (18)

## Data Availability

The data presented in this study are available in the article and Appendix A.

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
