# Peer review of "Treatment of High-Risk Neuroblastoma with Dinutuximab and Chemotherapy Administered in all Cycles of Induction"

_cancers, 2023, doi:10.3390/cancers15184609_

Round 1

Reviewer 1 Report

Very interesting and timely report as expanded use of anti-GD2 monoclonal antibodies with chemotherapy has continued apace!

 Worth comparing this experience (response, toxicity) with the hu14.18K322A (Furman et al, reference #5).

 Need to add some details about the cycles:

  • which day of a cycle was the dinutuximab started?
  • what was the duration of the infusion of dinutuximab?
  • was dinutuximab administered before, after, or simultaneously with the chemotherapy?
  • regarding IL-2:   what dosage, started on what day, what was duration of infusion?

With GM-CSF being administered for so many days, were there any problems with hypereosinophilia?

 Were cycles started about every 21 days?

 Among limitations (p.6 bottom). the authors touch on the issue of doing antibody in 1st cycle, but they still could provide rationale for starting antibody in the very 1st cycle   -   one might suppose the risk: benefit ratio is unfavorable as many newly-diagnosed patients have massive primary tumors or liver involvement that can entail respiratory and GI complications and antibody would not be expected to add much to cytoreduction of rapidly progressing disease in soft tissue disease or bones/bone marrow. 

 In Discussion, number of cycles and EOI responses are listed   -   could note that the single center report (ref #17) also included anti-GD2 antibody with the later cycles of induction in some patients, though not on same days as the chemotherapy.

This report presents so much more than just the experience doing the chemoimmunotherapy: a lot about subsequent treatments and outcome of the 25 patients   -   and most of p.7 addressing other treatments.   Not essential in this report, but well covered and therefore could remain.

Reviewer 2 Report

Overview

This is an extremely well written and clear presentation of a retrospective study of DIN administered to HRNBL patients during all cycles of induction therapy.  The results of a 96% end-of-induction response (PR) rate are very encouraging.  This reviewer agrees with the authors that further assessment of induction chemoimmunotherapy in a randomized clinical trial is warranted and has the potential to change the standard of care and outcomes.

Critique

The only concern, and it’s a big one, is that there was a bias introduced by selecting only certain HRNBL patients for treatment with DIN during all cycles of induction.  Such a bias is always the potential danger of a retrospective study.  How many patients diagnosed with HRNBL between 2017-2022 did not receive DIN during all cycles of induction and were excluded from the review?  If there were none, then great.  But if some of the HRNBL patients diagnosed between 2017-2022 did not receive DIN, then:

A. this should be discussed as a limitation of the study;

B. the authors should present the DIN and no DIN patients in a CONSORT diagram, including the possible reasons that the ‘no DIN’ group did not receive DIN.  If possible, a comparison of patient characteristics by DIN vs no DIN should be performed.  The purpose of this comparison should be clearly stated: that it is to check for bias of the DIN cohort, and not to test for superiority of response to DIN compared to no DIN.

Did the patients receive surgery?  That information should be added to the presentation of treatment and to Figure 1.

In the Results, it says,

“Two of 24 patients died from disease, including 1 patient who experienced PD while receiving therapy and 1 who relapsed 2 months after completing therapy. The median time off therapy of the 16 patients who completed therapy at time of publication and have not developed PD or recurrence is 9.8 months, ranging from 2.0 to 63.3 months.”

Does this mean that 6 patients developed PD or recurrence (i.e., 24-2=22; 22-16=6)?  If so, please add a sentence about this so that it’s clear where the 16 came from.

The authors make a convincing argument that 5 versus 6 cycles of induction therapy is unlikely to have resulted in a meaningful difference in outcome.

It is commendable that the authors addressed the possibility that the provision of housing, food, and transportation throughout the duration of therapy may have positively impacted results.

Reviewer 3 Report

Retrospective study of a selected cohort of patients

The main objective is to identify the response rate at the end of induction of a cohort of patients treated between 2017 and 2022.

One of the major question are the potential bias of the study

The criteria is "patients that received DIN combined with all Chemo cycles" which has been retrospectively identified and not "were planned to receive this treatment" . 

We have no information on the possible bias :

Number of patients treated during the same period in the institution. 

Criteria to be excluded from ongoing clinical trials

Patients treated outside clinical trials  but not included in this retrospective study

One patient is excluded of the analysis because of treatment stop because of toxicity at the begining of the 5th course (when 5 courses were planned according to the treatment flow chart) : why this patient is excluded and no characteristics given when it can be interpreted as a treatment failure

How many patients were planed to receive any of the 3 potential induction options described here and are not part of the study because of earlier stop (because of toxicity or PD)?

The results of a strategy cannot be analysed without taking into account the failures because of toxicity or early progression. 

Toxicity results : the toxicity is described globally : in the table (grade 3_4) the last column is the addition of the 2 previous ones which is useless;

On the other hand , it is stated in the discussion that the initial toxicity for newly diagnosd patients with bulky disease may be difficult to be managed in centers with less experience. There is no information in the results that support this point since the observed toxicity is not related to the cycle number and no specific problems faced initally by clinicians are reported . More specific information could be one of the added value of such a single center study.

The results are excellent which is very promising for all the "neuroblastoma community"

However the disease initial status and response is poorly reported.

It is reported by both COG and Siopen studies that  the MIBG score at diagnosis and at the end of induction are major prognosis factor . The metastatic response is now considered as a major prognosis factor. This point should be reported

Only the overal response is reported .

The MIBG score is not indicated neither at diagnosis nor at EOI which could be very informative to better understand what is this patient population and better identify the impact of the evaluated strategy, taking into account that the consolidation  heterogenity that cannot allow survival evaluation.

Discussion

The main study information  on response rate at the EOI is poorly discussed . The main part of the discussion are the very interesting authors' thoughts about perspectives of neuroblastoma treatment , which is a little irrelevant and should be shortened

Reviewer 4 Report

Very interesting and highly relevant since high risk NB continue to have a poor prognosis.

Heterogeneity of the sample and the fact that it is a retrospective study of a single center (already commented by the authors).

It would be useful to add the dinutuximab dose and administration scheme used.

This work invites a randomized and multicenter study, in addition to a long-term follow-up of patients. 

Round 2

Reviewer 3 Report

Most of the comments of my first review have been answered . 

The results are presented with the general background of the HR neuroblastoma patients treated in this institution during the study period, and the potential bias are discussed. 

the MIBG scores are still not reported which would have been a significant added value to this paper data  

the abstract and results chapters have been modified . 

One typo error raw 253: institutional